# Projected Incidence of Hepatobiliary Cancers and Trends Based on Age, Race, and Gender in the United States

**DOI:** 10.3390/cancers16040684

**Published:** 2024-02-06

**Authors:** Michael H. Storandt, Sri Harsha Tella, Mikolaj A. Wieczorek, David Hodge, Julia K. Elrod, Philip S. Rosenberg, Zhaohui Jin, Amit Mahipal

**Affiliations:** 1Department of Internal Medicine, Mayo Clinic, Rochester, MN 55905, USA; storandt.michael@mayo.edu; 2Department of Oncology, Mayo Clinic, Rochester, MN 55905, USA; tellas@slhs.org (S.H.T.); jin.zhaohui@mayo.edu (Z.J.); 3Department of Quantitative Health Sciences, Mayo Clinic, Jacksonville, FL 32224, USA; wieczorek.miko@mayo.edu (M.A.W.); hodge@mayo.edu (D.H.); 4Department of Statistics and Data Science, Carnegie Mellon University, Pittsburgh, PA 15213, USA; jkelrod97@gmail.com; 5Biostatistics Branch, Division of Cancer Epidemiology and Genetics, National Cancer Institute, Bethesda, MD 20814, USA; rosenbep@exchange.nih.gov; 6University Hospitals Seidman Cancer Center, Case Western Reserve University, Cleveland, OH 44106, USA

**Keywords:** hepatocellular cancer, gallbladder cancer, incidence, cholangiocarcinoma

## Abstract

**Simple Summary:**

Using historical data from the Surveillance, Epidemiology, and End Results database from 2001 to 2017, we were able to project the incidence of hepatobiliary malignancies between 2018 and 2029, and we expect rates of hepatobiliary malignancies to continue to increase during this time period. Specifically, rates of hepatocellular carcinoma and intrahepatic cholangiocarcinoma are expected to double from 2001 to 2029. The incidences of gallbladder carcinoma and ampullary cancer are expected to remain stable, while the incidence of extrahepatic cholangiocarcinoma is expected to increase. Certain age, racial, and gender subgroups are specifically expected to have higher rates of certain hepatobiliary malignancies in the coming years, including hepatocellular cancer and intrahepatic cholangiocarcinoma among whites, gallbladder cancer among blacks, and extrahepatic cholangiocarcinoma among Hispanics. Using this information, we can begin to allocate resources for screening, management, and research going forward, with a focus on specific cancer types and demographic groups.

**Abstract:**

Background: Identifying the projected incidence of hepatobiliary cancers and recognizing patient cohorts at increased risk can help develop targeted interventions and resource allocation. The expected incidence of subtypes of hepatobiliary cancers in different age groups, races, and genders remains unknown. Methods: Historical epidemiological data from the Surveillance, Epidemiology, and End Results (SEER) database was used to project future incidence of hepatobiliary malignancies in the United States and identify trends by age, race, and gender. Patients ≥18 years of age diagnosed with a hepatobiliary malignancy between 2001 and 2017 were included. US Census Bureau 2017 National Population projects provided the projected population from 2017 to 2029. Age–Period–Cohort forecasting model was used to estimate future births cohort-specific incidence. All analyses were completed using R Statistical Software. Results: We included 110381 historical patients diagnosed with a hepatobiliary malignancy between 2001 and 2017 with the following subtypes: hepatocellular cancer (HCC) (68%), intrahepatic cholangiocarcinoma (iCCA) (11.5%), gallbladder cancer (GC) (8%), extrahepatic cholangiocarcinoma (eCCA) (7.6%), and ampullary cancer (AC) (4%). Our models predict the incidence of HCC to double (2001 to 2029) from 4.5 to 9.03 per 100,000, with the most significant increase anticipated in patients 70–79 years of age. In contrast, incidence is expected to continue to decline among the Asian population. Incidence of iCCA is projected to increase, especially in the white population, with rates in 2029 double those in 2001 (2.13 vs. 0.88 per 100,000, respectively; *p* < 0.001). The incidence of GC among the black population is expected to increase. The incidence of eCCA is expected to significantly increase, especially among the Hispanic population, while that of AC will remain stable. Discussion: The overall incidence of hepatobiliary malignancies is expected to increase in the coming years, with certain groups at increased risk. These findings may help with resource allocation when considering screening, treatment, and research in the coming years.

## 1. Introduction

Hepatobiliary cancers encompass a spectrum of malignancies, including hepatocellular carcinoma (HCC), intra- and extrahepatic cholangiocarcinoma (iCCA and eCCA), gallbladder carcinoma (GC), and ampullary carcinoma (AC), and account for approximately 13% of cancer-related deaths globally [1]. The most common among these is primary liver cancer, including HCC and iCCA, which is the seventh most common cancer globally [2]. Other hepatobiliary malignancies are less common.

Approximately 41,000 new cases of primary liver cancer will be diagnosed in the United States (U.S.) in 2023, with 29,000 deaths [3], and liver cancer remains the third leading cause of cancer-related death globally [2]. From 2007 to 2016, the incidence of liver cancer in the U.S. increased [4,5]. HCC accounts for approximately 75% of primary liver cancers, with a majority of these cases attributed to modifiable risk factors, including chronic infection with hepatitis B or C, chronic ethanol use, and metabolic-associated steatohepatitis [6,7]. Unfortunately, liver cancer is associated with one of the lowest five-year survival rates at 21% [3], and as early detection is linked to better patient outcomes, identification of high-risk groups going forward is paramount. 

Extrahepatic biliary malignancies, including eCCA, GC, and AC, are less frequently seen, with an estimated incidence of 12,220 new cases in the U.S. in 2023, with 4510 deaths [3]. GC is the most common of these, accounting for 80–95% of extrahepatic biliary tract cancers, and is the 25th most common cancer globally, accounting for 1.2% of new cancer diagnoses [2,8,9]. Within the U.S., higher incidence is observed among females and the Native American population [9]. The incidence of eCCA has remained relatively stable in recent years, and risk factors for CCA include biliary flukes and hepatoliths in Southeast Asia and inflammatory conditions, such as primary sclerosing cholangitis, in the U.S. [1,10]. AC is the most rare, accounting for 0.2% of gastrointestinal malignancies [11].

Collectively, hepatobiliary malignancies convey a poor prognosis, and early detection is important in improving patient outcomes. Moreover, cancer treatment confers a high cost to the U.S. healthcare system and will continue to rise as newer, more expensive treatments emerge and the population ages. A study evaluating patient expenses related to cancer diagnosis estimated a national patient economic burden of $21.1 billion in 2019, with higher out-of-pocket costs experienced by those with advanced-stage disease [12]. As such, studies projecting incidence in future years play an important role in identifying demographics at risk for increasing incidence and allow for the allocation of resources for screening and prevention in high-risk populations. These interventions may then have the opportunity to reduce both symptom burden and financial toxicity experienced by the patient and the U.S. healthcare system. In the present study, we sought to project rates of hepatobiliary cancers in the U.S. over the next 5 years based on prior epidemiological data and evaluate change in incidence based on patient age, sex, and race. 

## 2. Methods

### 2.1. Incident Cases

We collected incidence and epidemiological data of hepatobiliary malignancy diagnoses between 2000 and 2017 from the Surveillance, Epidemiology, and End Results (SEER) registry database. The SEER database is a collection of cancer epidemiological data supported by the Surveillance Research Program in the National Cancer Institute’s Division of Cancer Control and Population Sciences and includes 30% of the newly diagnosed cancers in the U.S. We included patients ≥ 18 years of age who were diagnosed with HCC, iCCA, eCCA, GC, and AC (histology codes 8160–8163, 8140–8144, 8190, 8470/3, 8480, 8481, 8500–8508, and 8570–8576) during this time period. Patients who were <18 years of age, had more than 1 primary cancer location, were diagnosed on an autopsy report or on a death certificate only, or who had unknown or American Indian/Alaskan Native race due to low numbers in this group prohibiting incidence projection were excluded from this study.

### 2.2. Population Data

U.S. Census Bureau 2017 National Population projects provided the projected population from 2018 to 2029. The estimates are stratified based on age, sex, race, and Hispanic ethnicity. These projections are used in the calculation of the future estimates. 

### 2.3. Age–Period–Cohort (APC) Forecasting Models Analysis

The forecasting models we used were the APC (Age–Period–Cohort) models previously used by Rosenberg et al. [13]. These models were used to estimate future births in cohort-specific models. As performed previously, the estimates were obtained by multiplying the estimated longitudinal age incidence rate curve in the referent birth cohort by the rate ratio between birth-specific cohorts and the referent cohort. 

We age-standardized the incidence rates per 100,000 person-years using the 2010 US standard population. The projected absolute number of new cases was obtained by multiplying the projected incidence rates by age from the APC model by the projected population size from the U.S. Census Bureau. All analyses were completed by adapting the functions created by Rosenberg et al., using R Statistical Software (version 4.1.2; R Foundation for Statistical Computing, Vienna, Austria).

## 3. Results

We identified 110,381 patients diagnosed with hepatobiliary malignancy between 2000 and 2017 (Figure 1) after excluding patients who did not meet the inclusion criteria (Figure 1). The median age was 62.0 years; 69.6% were male, and 48.4% were non-white (Table 1). The most common diagnosis was HCC (68.9%), followed by iCCA (11.5%), GC (8.0%), eCCA (7.6%), and ampullary cancer (4.0%).

### 3.1. Hepatocellular Carcinoma 

Incidence of HCC is projected to continue to rise in the U.S. through 2029, with an anticipated incidence of 9.03 cases per 100,000 at that time, with males continuing to have a higher incidence than females (14.46 vs. 4.19 per 100,000) (*p* < 0.001) (Appendix A). This rate is nearly double that observed in 2001 (4.5 per 100,000). Rates of HCC are expected to increase most significantly in patients 70–79 years of age, with relative stability in younger cohorts (Figure 2a). This trend is observed in both females and males (Appendix A).

Among racial groups, a continued decline in incidence is anticipated among Asians, who accounted for the greatest incidence in 2001 (6.48 and 20.37 per 100,000 among females and males, respectively) and are projected to account for nearly the lowest incidence in 2029 (3.40 and 13.40 per 100,000 for females and males, respectively) (Figure 2b). Meanwhile, rates among Hispanic, black, and white patients are anticipated to increase, with Hispanics continuing to see the highest incidence among both females and males, followed by black and white patients. Increased rates among Hispanic, black, and white patients are expected primarily in the 70–79-year-old cohort for both females (Appendix A) and males (Appendix A).

### 3.2. Intrahepatic Cholangiocarcinoma

Cases of iCCA are anticipated to increase, with rates in 2029 double those of 2001 (2.13 vs. 0.88 per 100,000, respectively) (*p* < 0.001) (Appendix A). Overall incidence is comparable between females and males, with anticipated incidences of 2.15 and 2.16 per 100,000 in 2029. Increases are expected across races, with the exception of Asian and black males, among whom rates may stabilize (Figure 3a). While incidence among white females and males was near the lowest in 2001, by 2029, these groups may have the highest incidence among their respective sexes, with rates of 2.32 and 2.38 per 100,000, respectively (Figure 3a). By age, rates of increase are expected to be most significant in patients > 60 years of age (Figure 3b).

### 3.3. Gallbladder Carcinoma

The incidence of GC may remain relatively stable in the coming years, with an incidence of 0.88 per 100,000 in 2029, from 0.79 in 2001 (Appendix A). The incidence will be expected to remain higher in females compared to males (1.06 vs. 0.74 per 100,000, respectively) (Appendix A). The overall increase is anticipated to be primarily accounted for in the black population, with the projected incidence of 1.70 and 1.55 per 100,000 in females and males, respectively, in 2029, compared to an incidence of 1.21 and 0.32 in 2001, respectively (Figure 4a). Relative stability among age groups is suspected, with the highest incidence in the 70–79-year-old group (Figure 4b).

### 3.4. Extrahepatic Cholangiocarcinoma

The incidence of eCCA is expected to continue to increase in the near future, with 1.01 cases per 100,000 in 2029, compared to 0.59 in 2001 (*p* = 0.002). An increase is expected in both males and females, although eCCA remains more common in males, with the projected incidence of 1.27 and 0.79 per 100,000 in males and females, respectively, in 2029 (Appendix A). Increased rates are anticipated among all racial groups, with the exception of Asian females, with a projected decrease from 0.86 to 0.57 cases per 100,000 from 2001 to 2029, respectively (Appendix A). By 2029, Hispanic patients are expected to account for the highest incidence of eCCA in both females and males, with incidences of 1.41 and 1.57 per 100,000, respectively. An increase in incidence is primarily expected in patients >60 years of age (Appendix A).

### 3.5. Ampullary Carcinoma

AC is the rarest among hepatobiliary malignancies, with an incidence of 0.38 per 100,000 in 2001, projected to increase to 0.52 per 100,000 in 2029 (*p* = 0.12). AC is anticipated to increase in males, among whom this diagnosis is more common, and remain relatively stable among female patients (Appendix A). The incidence remains higher among Asian, black, and Hispanic individuals, with the most significant increase noted among black males (Appendix A). Incidence is expected to remain relatively stable across age cohorts, with the highest incidence noted in the 70–79-year-old population (Appendix A).

## 4. Discussion

Overall, the incidence of hepatobiliary malignancies is expected to increase in the coming years, and despite this, treatment options remain limited, and prognosis is often poor in patients with advanced diseases. In addition, the cost of cancer care continues to rise in the U.S., with increasing financial toxicity being experienced by patients with cancer. As such, it is essential that we recognize at-risk populations in order to allocate resources for screening and prevention, which may help mitigate the burden experienced by the patient and healthcare system. Additionally, projections may help guide research and drug development in areas of increasing need. In the above study, we were not only able to anticipate the incidence of hepatobiliary malignancies in the coming years but also identify novel trends among certain racial, gender, and age cohorts who may be at increased risk.

A limitation of this study is that these projections are based on prior trends and cannot take into account changes in screening, prevention, or risk factors that may develop in coming years. This study evaluated different demographic subpopulations with respect to age, race, and sex, and some subpopulations may have smaller sample sizes. Additionally, patients with more than one primary malignancy were excluded from analysis, which in this study was 30,000 patients, which may merit further exploration in the future. The SEER database includes 30% of the newly diagnosed cancers in the U.S. and is representative of the overall population but may not be easily generalizable to other countries. There is no concern for overdiagnosis as only confirmed cancer cases are included in the SEER database. The methodology used in our study is the established method to estimate future incidence and allows for the prediction and identification of groups who may require focused intervention. With respect to gallbladder cancer, patients who have undergone cholecystectomy are no longer at risk of developing gallbladder cancers, and our data implicitly estimated the gallbladder cancer risks in patients with intact gallbladders.

In the case of HCC, overall incidence is expected to double between 2001 and 2029, which is supported by a prior study [14]. This trend may be largely attributed to an increase in non-alcoholic steatohepatitis (NASH) in the setting of increasing rates of obesity and metabolic syndrome among the U.S. population, in addition to traditional risk factors such as alcoholic cirrhosis and chronic hepatitis [15,16,17]. Current guidelines are lacking with regard to screening for NASH, and increasing rates of HCC suggest a need to further evaluate the efficacy of screening in high-risk populations [18]. Additionally, it will be important to assess the impact of weight loss strategies for the management of NASH and its impact on rates of HCC. Historically, bariatric surgery has been commonly used for the management of obesity, and there are data that suggest an association between bariatric surgery and a lower incidence of HCC [19]. More recently, novel glucagon-like peptide-1 receptor (GLP-1) agonists have been developed and have shown significant benefits with weight reduction, including semaglutide and tirzepatide (also functions as a glucose-dependent insulinotropic polypeptide) [20,21]. Early evidence suggests that GLP-1 agonists may have benefits in the management of NASH [22,23]. As these medications become more common, it will be important to assess their impact on the incidence of HCC. Additionally, other avenues are being evaluated with regard to the treatment of NASH, which may also have an impact on the incidence of HCC [24].

Interestingly, we observe a continued decline in rates of HCC among the Asian population, which is supported by a prior study evaluating Asian–Americans but also by studies conducted in Asian countries [14,25,26]. This may be, in part, driven by public health efforts among Asian countries to reduce Hepatitis B infection through vaccination and screening and decrease exposure to aflatoxins, among other interventions [25,26]. As such, this may be reflected in U.S. incidence among Asian populations who migrate to the U.S. or are born in the U.S. without these exposures. Recently, the Centers for Disease Control and Prevention published a recommendation to expand hepatitis B screening to all adults above the age of 18 in the U.S. [27]. It will be important to address the implications of strategies such as this on controlling rates of HCC in the U.S., especially among the Asian population.

Among age groups, across all races with HCC, the most significant increase is expected among those 70–79 years of age. This population would have been born in the 1950s, and as Hepatitis C was not discovered until the 1980s, it is possible that this cohort may have been unknowingly exposed, as reflected in prior U.S. Preventive Services Task Force guidelines recommending hepatitis C screening in adults born between 1945 and 1965 (new guidelines recommend screening all adults 18–87 years) [28]. In the future, the treatment of hepatitis B and C may help in reducing the progression of liver disease to cirrhosis and subsequent development of HCC. 

Similar to HCC, cases of iCCA are expected to double from 2001 to 2029, with the most marked increase among white females and males, who had the lowest incidence in 2001 and may have the highest incidence in 2029. The overall increase in incidence is similar to previously observed trends [29]. However, the exact reasons for the rapidly increasing incidence among the white population remain unclear. Increasing recognition and accurate diagnosis may, in part, be responsible for increased incidence, as previously, iCCA was misclassified as cancer of unknown primary. In addition, there may be a relation to increasing rates of primary sclerosing cholangitis, which is most prominent in Northern European populations [30]. While rates are increasing in white patients, stability to moderate increase is seen in the Asian population. This trend may be attributed to public health efforts in Asian countries reflected in populations migrating to the U.S. or be related to rates of Asian individuals born in the U.S. without typical exposures, increasing risk for iCCA, including hepatitis B.

Similar to trends with iCCA, the incidence of eCCA is anticipated to increase, with a decline among Asian females and relative stability among Asian males. A continued increase is anticipated among Hispanic, black, and white patients, with the highest rates projected among Hispanic males and females. In spite of increasing rates of eCCA, an international study has found globally improved mortality in patients with eCCA, which has been largely attributed to advances in laparoscopic cholecystectomy [31].

The relative stability in the incidence of gallbladder carcinoma is anticipated, remaining infrequent, with an expected incidence of 0.88 cases per 100,000 in 2029. While stability is projected across racial groups, incidence is projected to increase in the black population, primarily in black males. Cholelithiasis is a primary risk factor for gallbladder carcinoma, with other risk factors including gallbladder polyps, metabolic syndrome, PSC, and heavy-metal or organochlorine exposure, among others [32]. The exact driver for increased rates among the black population in the U.S. is not evident and merits further evaluation.

## 5. Conclusions

Incidence of hepatobiliary malignancies is expected to continue to increase in the near future in the U.S. Hepatobiliary malignancies portends poor prognosis, especially in patients who are unresectable at the time of diagnosis, and it is important to identify groups at risk to develop targeted screening interventions. In the current study, we were able to identify novel trends in cancer incidence among different groups, including decreased incidence of HCC among Asian patients, increase in iCCA and HCC among white patients, and increase in GC among black patients. This may be useful in the development of targeted screening measures and allow for the focusing of resources in managing hepatobiliary cancers in the coming years. 

## Figures and Tables

**Figure 1 cancers-16-00684-f001:**
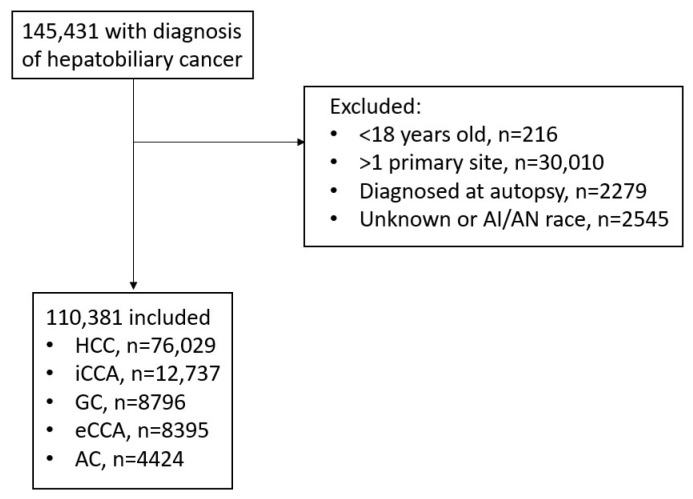
Consort diagram of patient cohort. Abbreviations: AI/AN, American Indian/Alaska Native; HCC, hepatocellular carcinoma; iCCA: intrahepatic cholangiocarcinoma; GC, gallbladder carcinoma; eCCA, extrahepatic cholangiocarcinoma; AC, ampullary carcinoma.

**Figure 2 cancers-16-00684-f002:**
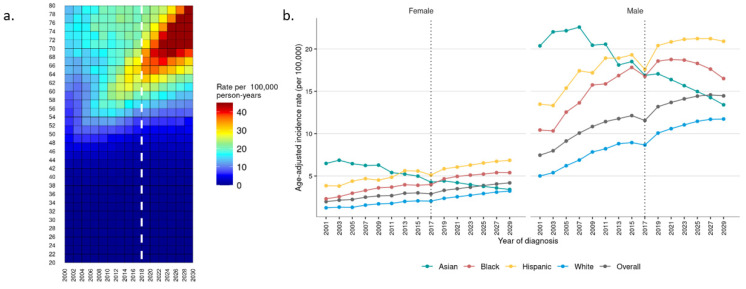
Observed (2001–2017) and projected (2018–2029) incidence of hepatocellular carcinoma by (**a**) age and (**b**) race in female and male patients.

**Figure 3 cancers-16-00684-f003:**
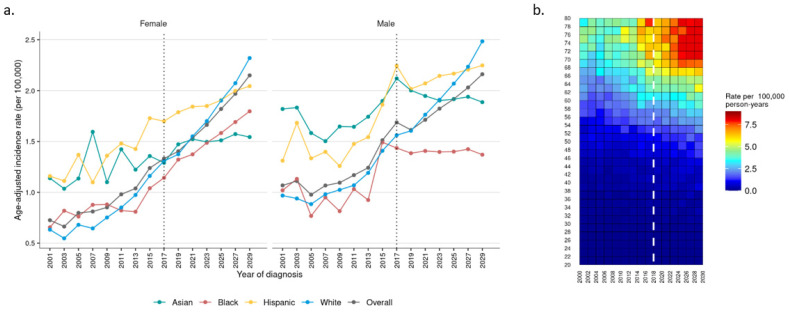
Observed (2001–2017) and projected (2018–2029) incidence of intrahepatic cholangiocarcinoma by (**a**) race for females and males and (**b**) age.

**Figure 4 cancers-16-00684-f004:**
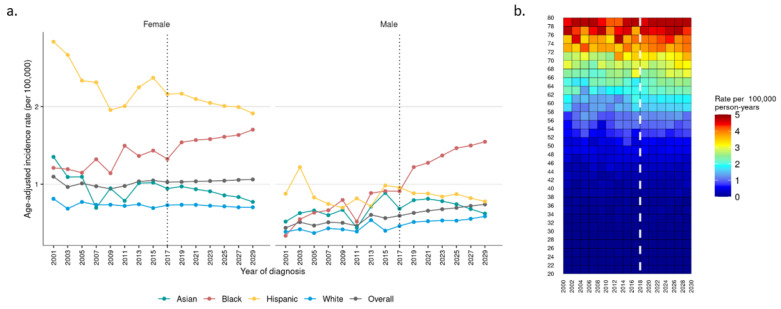
Observed (2001–2017) and projected (2018–2029) incidence of gallbladder carcinoma by (**a**) race in females and males and (**b**) age.

**Table 1 cancers-16-00684-t001:** Demographics and primary tumor location of entire cohort. Abbreviations: HCC, hepatocellular carcinoma; iCCA, intrahepatic cholangiocarcinoma; GC, gallbladder carcinoma; eCCA, extrahepatic cholangiocarcinoma; AC, ampullary carcinoma.

	All (*n* = 110, 381)
**Age at diagnosis, median (range)**	62.0 (20.0, 79.0)
**Sex**	
Male	76,847 (69.6)
Female	33,534 (30.4)
**Race/Ethnicity**	
White	56,955 (51.6)
Hispanic	22,158 (20.1)
Asian	16,813 (15.2)
Black	14,455 (13.1)
**Cancer type**	
HCC	76,029 (68.9)
iCCA	12,737 (11.5)
GC	8796 (8.0)
eCCA	8395 (7.6)
AC	4424 (4.0)
**Primary location listed**	
Liver	79,614 (72.1)
Intrahepatic bile duct	9152 (8.3)
Gallbladder	8796 (8.0)
Extrahepatic bile duct	8395 (7.6)
Ampulla of Vater	4424 (4.0)

## Data Availability

Publicly available datasets were used, and specific datasets can be requested by contacting the corresponding author.

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
