# Peer review of "Projected Incidence of Hepatobiliary Cancers and Trends Based on Age, Race, and Gender in the United States"

_cancers, 2024, doi:10.3390/cancers16040684_

Round 1
Reviewer 1 Report
Comments and Suggestions for Authors
This is an important study on a cutting edge topic.
I just would like to emphasize the important of therapies for NASH on HCC incidence. The authors mentioned a drug but there are many other pharmacological agents for NASH currently under investigation....maybe a table could be of help in this case. The authors should mention also the beneficial role of other non-pharmacological therapies for NAFLD/NASH, such as bariatric surgery or endoscopy.
Author Response
I just would like to emphasize the important of therapies for NASH on HCC incidence. The authors mentioned a drug but there are many other pharmacological agents for NASH currently under investigation....maybe a table could be of help in this case. The authors should mention also the beneficial role of other non-pharmacological therapies for NAFLD/NASH, such as bariatric surgery or endoscopy.
Reply: Thanks for the suggestion. We have expanded upon this section, noting more specific details about various management strategies for NASH, including GLP-1s and bariatric surgery.
Reviewer 2 Report
Comments and Suggestions for Authors
Interesting and well conducted study on a large sample of patients.
The authors should expand their comment on the incidence of NASH on projected occurrence of HCC, particularly mentioning the preventive measure against this aspect. THey properly cite a pharmacological drug but they should also mention the impact of bariatric surgery on NASH-related HCC incidence (cite the recent SRMA PMID: 33721336 )
Author Response
The authors should expand their comment on the incidence of NASH on projected occurrence of HCC, particularly mentioning the preventive measure against this aspect. THey properly cite a pharmacological drug but they should also mention the impact of bariatric surgery on NASH-related HCC incidence (cite the recent SRMA PMID: 33721336 )
Reply: We have expanded upon this section and now note the role of bariatric surgery in reducing incidence of HCC as per the suggestion.
Reviewer 3 Report
Comments and Suggestions for Authors
Dear authors
The manuscript you is very well written, as well as the images display all the information.
I would like to congratulate you for this paper.
Author Response
The manuscript you is very well written, as well as the images display all the information.
I would like to congratulate you for this paper.
Reply: We appreciate this feedback.
Reviewer 4 Report
Comments and Suggestions for Authors
This is an interesting study in the era of hepatobiliary cancers, examining both the present status and trends in future incidence, in US population. As authors acknowledge, a major limitation is that parameters affecting incidence, mainly changes in the epidemiology of risk factors and advances in their treatment, were not included in the calculation of the estimations. However, the study offers valuable and alarming information about future, if current epidemiology and practice remained unchanged.
Major comments:
· The manuscript needs extensive editing for grammar and syntax errors.
· In my opinion, authors should clearly state both in title and abstract that the study was conducted in USA, so that the results cannot be generalized in other populations with different epidemiology of well-known risk-factors for hepatobiliary malignancies and diverse surveillance programs. They should also make a comment in discussion.
· Authors should clarify in “methods” the proportion of US population that is included in SEER database.
· The incidence of synchronous malignancy is estimated at 4.5-11.7% in the literature [Demandante CGN et al. Am J Clin Oncol 2003 Feb;26(1):79-83]. According to figure 1, there were 30.000 patients with more than one primary malignancy, leading to an incidence of about 20% in the present manuscript. Authors should make a comment about this. Furthermore, it would be of great interest if authors could provide more information about this group, specifically, types of synchronous malignancies.
Comments on the Quality of English LanguageThe manuscript needs extensive editing for grammar and syntax errors
Author Response
The manuscript needs extensive editing for grammar and syntax errors.
Reply: We have closely reviewed for any grammatical errors.
In my opinion, authors should clearly state both in title and abstract that the study was conducted in USA, so that the results cannot be generalized in other populations with different epidemiology of well-known risk-factors for hepatobiliary malignancies and diverse surveillance programs. They should also make a comment in discussion.
Reply: We have changed the title and abstract to reflect this and have made a comment in the discussion section.
Authors should clarify in “methods” the proportion of US population that is included in SEER database.
Reply: We have added a comment to the methods section that the SEER database includes 30% of newly diagnosed cancers in the United States.
The incidence of synchronous malignancy is estimated at 4.5-11.7% in the literature [Demandante CGN et al. Am J Clin Oncol 2003 Feb;26(1):79-83]. According to figure 1, there were 30.000 patients with more than one primary malignancy, leading to an incidence of about 20% in the present manuscript. Authors should make a comment about this. Furthermore, it would be of great interest if authors could provide more information about this group, specifically, types of synchronous malignancies.
Reply: Thanks for the suggestion. We have made a comment in the discussion regarding exclusion of these patients from the current manuscript. Unfortunately, analysis regarding details of this subpopulation is not available at this time. Interpretation of data from patients with synchronous malignancies is challenging and is complicated by the fact that these patients may be misdiagnosed with having primary liver cancer versus liver metastases. In the previous publications in different malignancies, this group is historically excluded.
Round 2
Reviewer 1 Report
Comments and Suggestions for Authors
The revised manuscript is OK. Thank you!
Reviewer 2 Report
Comments and Suggestions for Authors
It is ok. Thank you
Reviewer 4 Report
Comments and Suggestions for Authors
I have no further comments to make
Comments on the Quality of English LanguageLine 289: Please rephrase " in patients who are unresectable"